# Drying Characteristics of *Eucalyptus urophylla × E. grandis* with Supercritical CO_2_

**DOI:** 10.3390/ma13183989

**Published:** 2020-09-09

**Authors:** Jing-Wen Zhang, Hong-Hai Liu, Hang Yang, Lin Yang

**Affiliations:** 1College of Furnishings and Industrial Design, Nanjing Forestry University, Nanjing 210037, China; Wennyzhang96@163.com (J.-W.Z.); yh792503489@163.com (H.Y.); 2Co-Innovation Center of Efficient Processing and Utilization of Forest Resources, Nanjing Forestry University, Nanjing 210037, China; 3Key Laboratory of Bio-based Material Science and Technology (Northeast Forestry University), Ministry of Education, Harbin 150040, China

**Keywords:** *Eucalyptus urophylla × E. grandis*, supercritical CO_2_ drying, drying characteristic, collapse, drying stress

## Abstract

Supercritical CO_2_ (ScCO_2_) is a drying medium with excellent solubility and mass transfer efficiency. Supercritical CO_2_ drying (SCD) can remove the water of wood rapidly and prevent a change of microstructure caused by capillary tension in the drying process. In this study, *Eucalyptus urophylla × E. grandis* specimens with lengths of 50 and 100 mm were dried with ScCO_2_. Conventional kiln drying (CKD) and oven-drying (OD) were used as control. After 1 h, the drying rate, shrinkage, moisture distribution, drying stress were measured to explore the influence of drying methods and specimen length for drying characteristics during the early drying stage. The results showed that compared with CKD and OD, water removal was the fastest under SCD, and the drying rate was nine times of CKD and one time of OD. The shrinkage of SCD was the lowest among the three drying methods. Moisture distribution of SCD and OD was uneven. The drying stress of SCD was relatively high, the drying stress index of it was almost five times of CKD and three times of OD. Regardless of the drying method, shorter specimens had a shorter drying period but greater drying defects than the long specimens.

## 1. Introduction

With the rapid development of China’s forest product industry, the demand for wood resources is increasing. However, China’s limited natural forest resources cannot meet the demand of the forest product industry, and the imbalance of supply and demand leads to China’s wood resources depending on imports [1]. Planted forests have a fast growth rate and high survival rate, and their development and planting can bring abundant wood resources to China’s forest production industry [2].

Eucalyptus is one of the main tree species in China’s planted forests, with high yield, short rotation cutting cycle, multiple varieties, hard texture, and beautiful patterns. Eucalyptus is also an important raw material for the production of wood-based panels, wood floors, and wooden furniture [3]. However, due to a large number of extracts, poor permeability, large growth stress, cracking, deformation, warping, shrinkage, and other drying defects are easy to occur during conventional kiln drying (CKD), which results in a decrease of timber strength and timber utilization ratio [4]. The key technology to reduce the drying defects mentioned above is to develop suitable drying methods.

Currently, the CKD method is the most commonly used schedule for the eucalyptus industry. However, CKD has some drawbacks, such as a long drying cycle, high drying stress, and environmental pollution. Many scholars also use special drying methods to dry eucalypti, such as vacuum drying, microwave drying, freeze-drying, and solar drying [5,6,7,8]. Compared with the mentioned above drying methods, supercritical CO_2_ drying (SCD) has the advantages of fast drying rate, environmental protection, and simple operation [9]. ScCO_2_ is a fluid that exceeds the critical pressure of CO_2_ (7.38 MPa) and the critical temperature (31.1 °C). It has a similar density as liquid and similar viscosity as the gas, thus it has good solubility and heat transfer capability, and thereby can remove water with maximum extent without changing the structure of the cell wall [10,11,12].

Eucalyptus is prone to collapse [13,14]. The collapse of eucalyptus is attributed to the negative pressure of liquid capillary tension to crush cell walls when water exits the micropores in the cell walls at the initial stage of drying [15,16,17,18,19]. Collapse results in abnormal deformation of Eucalyptus cells at the stage of high moisture content (MC) [20]. Specific drying methods can reduce the collapse, such as steaming pretreatment [21], freeze-drying [22], supercritical CO_2_ drying [9,15]. Due to the lumen water expulsion during SCD, the microstructure changes caused by capillary tension can also be reduced, which plays a certain role in preventing wood collapse [15].

A previous study [9] has investigated the effects of ScCO_2_ pressure, ScCO_2_ temperature, and hold time on the drying rate of SCD. Compared to temperature and hold time, pressure has the greatest influence on the drying rate. However, there is a lack of discussion about the influence of SCD on the other wood characteristics, such as shrinkage and drying stress, which have a significant impact on wood follow-up processing quality. Dawson and Pearson [23] studied SCD, OD, and SCD combined OD to dry softwood and hardwood. SCD combined OD would benefit the most hardwood heartwoods in collapse, shrinkage, and checking. There is still no comparative study on CKD which is the most commonly used wood drying method. Behr et al. [24] and Newman et al. [11] used proton magnetic resonance imaging (MRI) to study the moisture distribution of SCD. The strength of the signal in the image to represent changes of MC before and after wood drying and the obvious difference of MC from earlywood and latewood. However, the MRI shows only the relative moisture distribution, not the specific. In this study, the weighing method and contour chart were used to describe the moisture distribution and detailed MC on the wood. Many studies [25,26] showed that specimen dimensions would lead to different free water transfer in the wood, which would have an impact on wood drying quality. The previous study [27] has investigated the influence of specimen thickness on SCD. The study shows the final MC of SCD was independent of specimen thickness There is no study on the influence of specimen length on SCD.

CKD and OD are the most common drying methods. In this paper, we proposed a hypothesis that SCD can benefit wood drying quality (drying rate, collapse, drying stress) regardless of different specimen length by comparing CKD and OD. The experiment was carried on the initial stage of drying because the collapse occurs at the high MC stage. We evaluated drying rate, moisture distribution, drying stress index and shrinkage rate to explore the influence of drying methods and specimen length on the *Eucalyptus urophylla × E. grandis* wood during the stage of initial drying, to provide a reference for improving the drying quality and expanding the solid wood utilization of eucalyptus.

## 2. Materials and Methods

### 2.1. Materials

*Eucalyptus urophylla × E. grandis* logs were produced in Liuzhou, Guangxi province, China, with the initial MC of 124.4% and the basic density of 0.502 g·cm^−3^. The logs were processed into lumbers with specifications of 30 (*Transversal*) × 25 (*Radial*) × 1000 mm (*Longitudinal*), wrapped with plastic film, and stored in the freezer for later use. The lumbers were sawed into specimens with lengths of 50 and 100 mm, as shown in Figure 1. The specimens were randomly divided into three groups, for SCD, CKD, and OD separately. Eighteen specimens were used for one particular drying method, including nine 100 mm specimens and nine 50 mm specimens. For one length, six specimens were used for shrinkage measurement, three specimens for the measurement of drying rate, moisture distribution, and drying stress. Three replicates were used for each measurement.

### 2.2. Supercritical CO_2_ Drying

The SCD system (DY221–50–06, Nantong HuaAn Supercritical Fluid Extraction Co., Ltd., Nantong, China) was composed of a CO_2_ bottle, 5 and 2 L drying vessel, circulating pump, and drying adsorption vessel, as shown in Figure 2 and Figure 3. Table 1 shows the parameters of SCD.

Supercritical CO_2_ drying steps: (1) Weighted specimens were put into the drying vessel and then the vessel was sealed. (2) The vent valve of the CO_2_ bottle was opened and the liquid CO_2_ was pumped into the drying vessel. (3) The CO_2_ was converted into ScCO_2_ by increasing pressure and temperature. (4) Pressure (30 MPa) and temperature (45 °C) were maintained during the hold time (15 min) at the drying vessel. (5) During the drying process, the specimens were fully contacted with ScCO_2_. (6) After 15 min, the pressure decreased to atmospheric pressure. At the same time, ScCO_2_ converted into CO_2_ bubbles, which carried moisture away from the wood. (7) The specimens were taken out from the vessel and weighed after CO_2_ had been exited from wood. (8) The specimens were returned to the vessel. Cycles (1)–(7) were repeated four times. The total drying time is 1 h, and the specimens were at the initial stage of drying. (9)The slices were cut from the specimens for moisture distribution, shrinkage, and drying stress (Figure 1) after the last cycle.

### 2.3. Conventional Kiln Drying and Oven-Drying

The CKD was carried according to the previous schedule [28], which ensures that the *Eucalyptus urophylla × E. grandis* has a beneficial drying quality under the 50 °C dry-bulb temperature and 84% relative humidity. The specimens were taken out from the constant temperature and humidity chamber (DF–408, Nanjing FuDe Instrument Co., Ltd., Nanjing, China) every 15 min to measure the weight for MCs calculation. After 1 h drying, the specimens had been weighed four times in the initial stage of drying, the time–MC curve was drawn after MCs calculation. The three test slices were cut from the specimens for moisture distribution, shrinkage, and drying stress (Figure 1).

OD was carried under the condition of 100 °C in the drying oven (DHG–905386–III, Shanghai Cimo Medical Instrument Co., Ltd., Shanghai, China) for 1 h. The measurements were the same as CKD.

### 2.4. Moisture Content

The MCs during and after drying were determined according to the China National Standard (GB/T 1931–2009) [29]. The MC specimens were put into an oven and dried at (103 ± 2) °C. Specimens were taken out and weighed every 2 h until the difference between the weights is no more than 0.5%. At this point, the specimens were considered absolute drying. The MCs were calculated according to Equation (1).
*W* = (*m*_1_ − *m*_0_)/*m*_0_ × 100%(1)
where *W* is the MC (%), *m*_1_ is the quality of the specimens when tested (g) and *m*_0_ is the quality of the absolute drying specimens (g).

### 2.5. Shrinkage

The shrinkage was measured by the scanning method [16]. The shrinkage slices with 2 mm thickness were cut from the center of the specimens before drying (Figure 1), and scanned by the scanner (CanoScan LiDe 700F, Canon Inc., Ho Chi Minh, Vietnam), the images (Image resolution: 300 dpi; Bit depth: 24) of initial shrinkage slices were obtained. The complete specimens were used during dying. After drying, the shrinkage slices with 2 mm thickness were also cut from the center of the complete specimens (Figure 1) and scanned, the images of final shrinkage slices were obtained. Adobe Photoshop CC 2019 (Adobe Systems Inc., San Jose, CA, USA) was used to measure the pixel of the shrinkage specimens in the scanning images. The shrinkage rate was calculated using Equation (2).
*β* = (*N*_0_ − *N*_1_)/*N*_0_ × 100%(2)
where *β* is the shrinkage rate (%), *N*_0_ is the pixel of the initial shrinkage slices before drying in scanning images (px), and N_1_ is the pixel of final shrinkage slices after drying in scanning images (px).

### 2.6. Moisture Distribution

The moisture distribution slices with 5 mm thickness were cut from the central part of specimens (Figure 1). The distance from the slices to the end of specimens is (22 ± 0.5) mm in 50 mm specimens and (47 ± 0.5) mm in 100 mm specimens. Then, the moisture distribution slices were cut into 25 small blocks uniformly (Figure 1) [9,30]. The MCs measurement of each small block was the same as Section 2.4 (Moisture Content). The moisture distribution images were drawn by Origin 2020 (OriginLab Corp., Northampton, MA, USA).

### 2.7. Drying Stress

The drying stress measurement is according to the China National Standard (GB/T 6491-2012) [31]. The drying stress slices with 10 mm thickness were cut from specimens after drying. The distance from the slices to the end of specimens is 5 mm in both specimens of 50 and 100 mm. Then, as shown in Figure 1, the slices were cut into a concave shape along the cutting lines. The initial teeth width *S* and length *L* were measured using an electronic digital caliper (distinguishability = 0.01 mm, precision = ±0.02 mm) after cutting. After that, the slices were placed at room temperature for 24 h in a well-ventilated area. Then, the dimension of final teeth width *S*_1_ was measured using an electronic digital caliper. The drying stress index *Y* was calculated using Equation (3).
*Y* = (*S* − *S*_1_)/2*L* × 100%(3)
where *Y* is the drying stress index (%), *S* is the initial teeth width of slices (mm), *S*_1_ is the final teeth width of slices (mm), and *L* is the teeth length of slices (mm).

## 3. Results and Discussion

### 3.1. Drying Rate

The time–MC curves of specimens under different drying conditions were shown in Figure 4, and the drying rate values were calculated in Table 2. The MC of the specimens decreased differently under the three drying methods. The slope of the curves represents the drying rate, showing that SCD had the highest drying rate, followed by OD, while CKD had the lowest rate. The highest drying rate was attributed to the dewatering property of ScCO_2_. The SCD process is composed of pressurization, holding, and depressurization. Free water in wood became more easily dissolved in ScCO_2_ after the pressure rose to 30 MPa [32]. A great volume of CO_2_ gas bubbles generated and expelled water out of wood as the pressure was lowered. Besides, the pressure difference also broke through the occluded pit of wood, thereby improving the permeability which accelerated the drying rate of SCD [9]. Combined with the values in Table 1, it can be concluded that the drying rate of SCD was approximately nine times that of CKD and one times that of OD. The temperature of OD was 100 °C, higher drying temperature accelerated the moisture loss and faster drying rate. To ensure a small cracking and deformation of wood, CKD at a lower temperature and higher humidity, such drying conditions led to slow moisture loss, thereby reducing the drying rate among the three drying methods.

Compared with 100 mm specimens, the drying rates of 50 mm were improved by 3.8%, 29.7%, and 34.3% in SCD, CKD, and OD separately. These findings suggested that the drying rate was affected by the length of specimens, and there was little impact on SCD. For CKD and OD, free water was removed more easily from interior to surfaces in short specimens. However, for SCD, free water was removed almost the same from interior to surfaces in both 50 and 100 mm specimens due to bubbles forcing the water out of the wood.

The two-way (drying method and specimen length) analysis of variance with drying rate was shown in Table 3. The drying method and specimen length had a significant influence on the drying rate (*p* < 0.05). The interaction between these two factors was also significant (*p* < 0.05).

### 3.2. Shrinkage

Figure 5 shows the shrinkage rate of specimens with lengths of 50 and 100 mm after 1 h drying of three different drying methods. For specimens under the initial drying stage in this study, the shrinkage of wood is dominated by collapse. Therefore, the collapse of wood is represented by the shrinkage rate.

It can be seen in Figure 5, the largest shrinkage rate was OD, followed by CKD and the smallest was SCD. The shrinkage rate of SCD was the smallest which was similar to a previous report [9]. This is due to the free water expelled from wood by CO_2_ bubbles which can reduce capillary tension during dewatering, hence preventing the increase of negative pressure of water and the collapse of eucalyptus [15]. The shrinkage rate of CKD was attributed to the moderated temperature and high relative humidity. For OD, the collapse was caused by great liquid capillary tension due to the fast water evaporation and migration at high temperatures.

The shrinkage rate of the short specimens in three drying methods was higher than the long specimens. This is probably because of the faster migration of free water in the short specimens, which increased the capillary tension and the degree of cell collapse.

It is shown in Table 4 that the drying method was significant (*p* < 0.05) on the shrinkage rate, but the length of the specimens had no significant effect (*p* > 0.05). The interaction between the two factors was not significant (*p* > 0.05). Compared with the length of the specimen, the drying method has a greater influence on the shrinkage rate.

### 3.3. Moisture Distribution

Figure 6 shows the moisture distribution image of the specimens with lengths of 50 and 100 mm under three drying methods. The figures show that the moisture distribution of specimens with SCD and OD were uneven, and the MC gradient between the core zone and surface zone was large. While, in CKD, the moisture distributed was even. This is because the drying rate of SCD and OD is faster than CKD, the rapid migration of free water led to the faster decrease of surface water, and the interior free water could not transfer to the surface zone in time. The distribution of SCD presented was more irregular compared with OD which was similar to a ring. This was attributed to the special driving force of water removal. Free water during SCD was removed in the shortest path in the wood by the force of bubbles of CO_2_ during the depressurization. However, the CKD condition was low temperature and high humidity, which inhibited the evaporation rate of water in the surface zone, thus the moisture distribution was more even.

The different final MC of the three methods also had an influence on moisture distribution. Above FSP, the moisture distribution became more evenly as MC decreased during SCD in the previous study [9]. For OD, the MC of the core part was high and the surface part was low, this MC gradient was always present above the FSP, while this gradient was reduced with the decrease of MC [33]. The moisture distribution was almost even in the different MC above FSP during CKD [34]. Table 5 shows the final MC of three drying methods variance analysis on moisture distribution. The final MC of SCD had a significant effect (*p* < 0.05), while the CKD and OD had no significant effect (*p* > 0.05).

No matter which drying method was used, the short specimens compared with the long specimens had a similar moisture distribution (the core has high MC while the surface has low MC in SCD and OD, even moisture distribution in CKD), suggesting that the different length of the specimens had little influence on moisture distribution. In Table 6, the drying method had a significant influence (*p* < 0.05) on moisture distribution while specimen length had no significant influence (*p* > 0.05). Combined with Table 5, the moisture distribution of SCD was affected by the drying method and final MC, while OD and CKD were only affected by the drying method.

### 3.4. Drying Stress

Figure 7 shows the drying stress index of three drying methods with lengths of 50 and 100 mm. The drying stresses all were lower than 1% and had small changes in three drying conditions. This is because few drying stresses occurred when MC above FSP. The uneven MC distribution and MC gradients provided the driving force for stress development [35,36,37]. As reported by a previous study [38], the greater MC gradient within wood resulted in greater stress and deformation of wood during high MC stages. Thereby, the relative higher drying stress of SCD in this study may be attributed to the greater MC gradients. In addition, the pressure gradient within wood may also affect the stress development of wood during SCD. It also can be seen from Figure 7 that the drying stress of 50 mm specimens was larger than 100 mm specimens, this also can be explained by the greater MC gradients in short specimens. However, the final MC distribution of CKD was even, thus the drying stress was small. Table 7 shows the drying method was significant (*p* < 0.05) on the drying stress, but the length of the specimens had no significant effect (*p* > 0.05). The interaction between the two factors was not significant (*p* > 0.05).

## 4. Error Analysis

Table 8 shows the error analysis of the experiment data, using the sum of the squared errors (SSE) and standard deviation (SD) to show the dispersion of the data. The value of SSE and SD is small in the table, which proves the reliability of the data.

Since CO_2_ continues to be discharged from the wood after SCD, this would cause test error. However, the waiting time for CO_2_ discharge after drying can be controlled to reduce this error. In the measurement of MC, the MC of small blocks of wood will change dramatically after being cut if they are not weighed timely. Before being cut again, the other small wooden blocks were put in a sealed bag to prevent MC changes to reduce error.

## 5. Conclusions

In this study, *Eucalyptus urophylla × E. grandis* specimens with lengths of 50 and 100 mm were dried with ScCO_2_. CKD and OD were used as control. The drying rate, shrinkage rate, moisture distribution, and drying stress index were measured to explore drying characteristics in different drying methods and specimen length. Based on the test, the following conclusions were drawn:In the wood initial drying stage, compared with CKD and OD, the SCD had the highest drying rate. After drying for 1 h, the MC of specimens dropped very rapidly by SCD, the MC of the short specimens decreased from 108% to 51%, and the long specimens decreased from 112% to 57%. The drying rate of SCD was nine times that of CKD and one time of OD.Due to its strong solubility and transfer capability, ScCO_2_ can reduce the capillary tension, thus achieving the effect of inhibiting wood collapse. The shrinkage rate of short and long specimens was 0.855% and 0.191%, respectively, both lower than the values of CKD and OD.However, the drying stress generated by SCD was relatively higher due to the uneven moisture distribution. The drying stress index of short specimens and long specimens was 0.59% and 0.38%, five and three times of CKD and OD, respectively.Regardless of the drying method, due to the shorter free water migration distance and faster migration speed, the short specimens had the following rules compared with the long specimens: a faster drying rate, greater shrinkage rate, and greater drying stress. That is, shorter specimens had a shorter drying period but greater drying defects than the long specimens.

## Figures and Tables

**Figure 1 materials-13-03989-f001:**
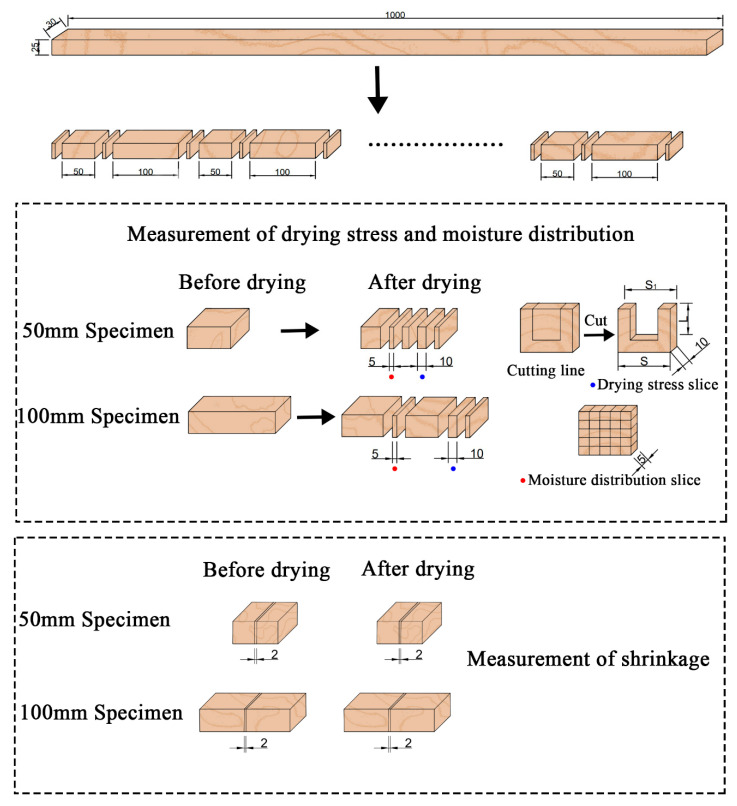
Diagram of specimens cutting.

**Figure 2 materials-13-03989-f002:**
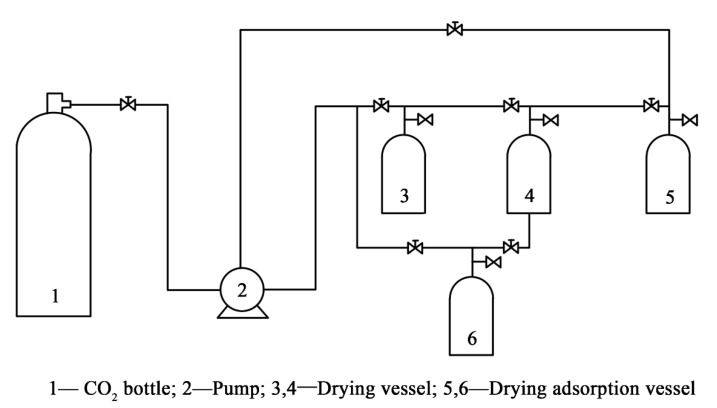
Supercritical CO_2_ drying system.

**Figure 3 materials-13-03989-f003:**
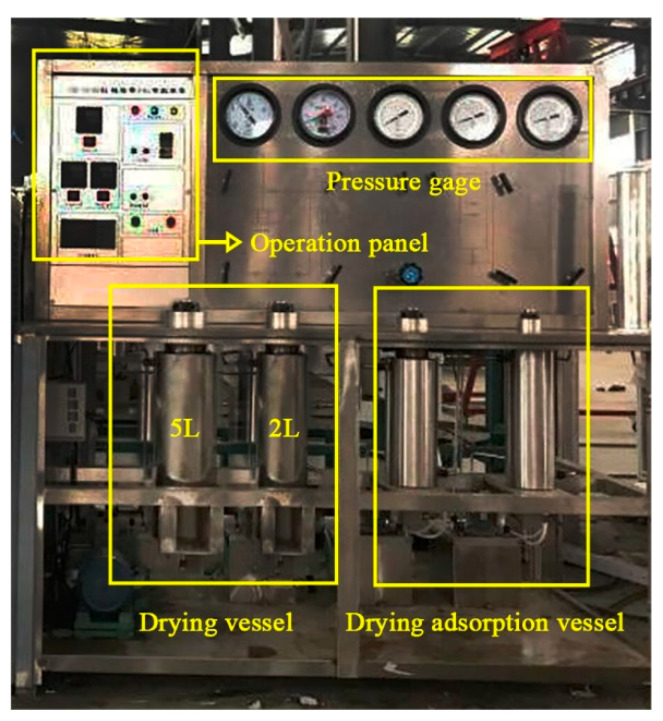
Supercritical CO_2_ drying instrument onsite.

**Figure 4 materials-13-03989-f004:**
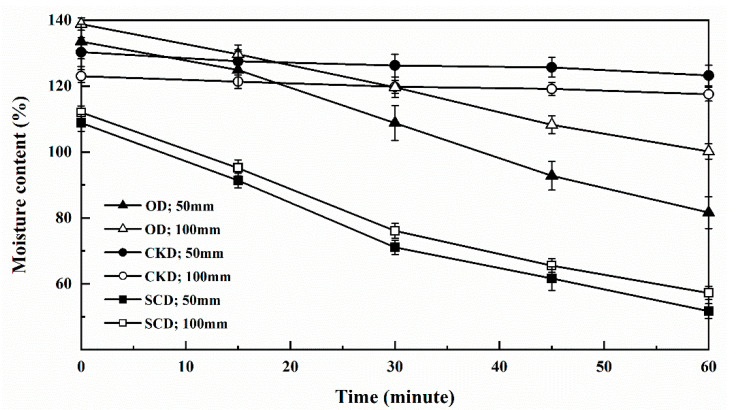
Curves of moisture content (MC) against the time of specimens of different drying methods.

**Figure 5 materials-13-03989-f005:**
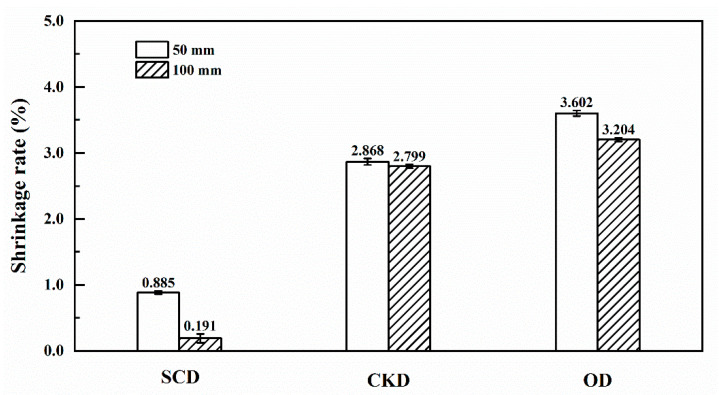
Shrinkage rate of specimens of different drying methods.

**Figure 6 materials-13-03989-f006:**
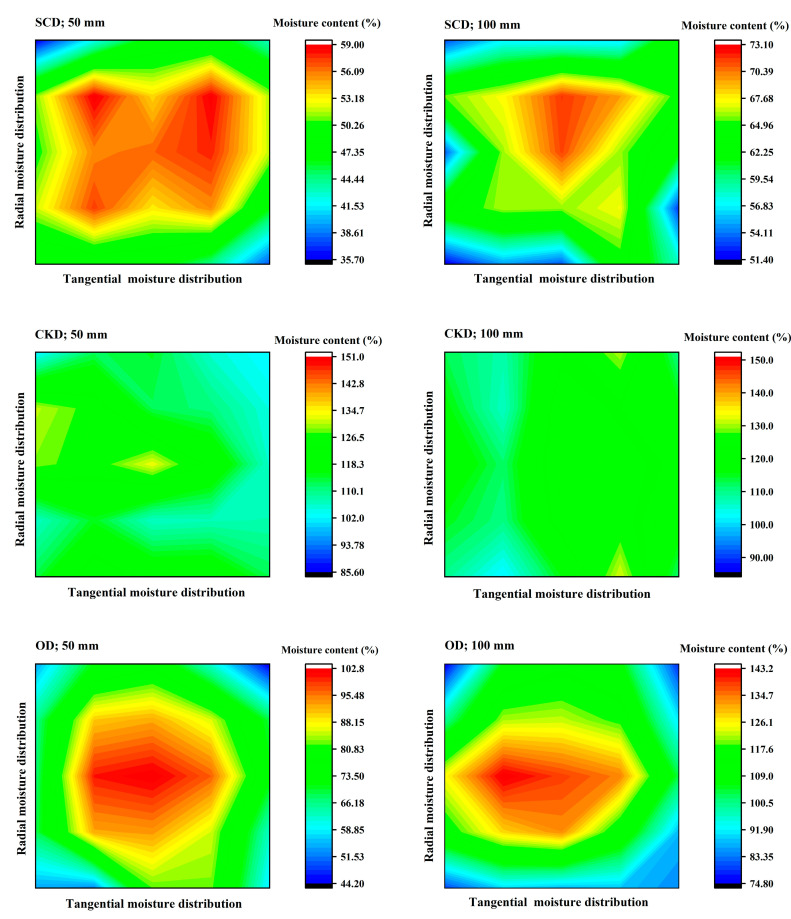
Moisture distribution of specimens under different drying conditions.

**Figure 7 materials-13-03989-f007:**
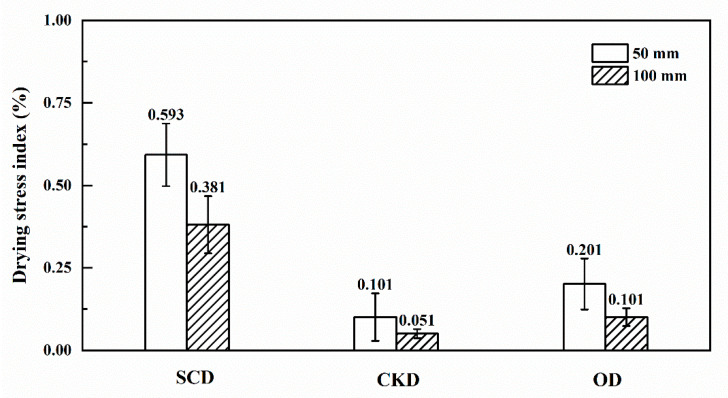
Drying stress of specimens under different treatment conditions.

**Table 1 materials-13-03989-t001:** Parameters of supercritical CO_2_ drying.

Process Parameter	Value
Maximum pressure (MPa)	30
Minimum pressure (MPa)	0.1 (atmospheric pressure)
Pressurization time (min)	30
Depressurization time (min)	10
Temperature (°C)	45
Hold time (min)	15
Number of cycles	4

**Table 2 materials-13-03989-t002:** Drying rate of specimens under different drying methods.

Drying	Specimen Length (mm)	Initial MC (%)	Final MC (%)	Drying Time (min)	Drying Rate (%·min^−1^)
SCD	50	108.8	51.7	60	0.952
	100	112.2	57.2	60	0.917
CKD	50	130.3	123.2	60	0.118
	100	123.0	117.5	60	0.091
OD	50	133.5	81.6	60	0.865
	100	138.8	100.1	60	0.644

**Table 3 materials-13-03989-t003:** Two-way analysis of variance with drying rate.

Source	DF	Sum of Squares	Mean Square	F Value	*p* Value
Factor A (Drying method)	2	1.6497	0.8249	832.6613	4.6267 × 10^−8^
Factor B (Specimen length)	1	0.0482	0.0482	48.6187	4.3255 × 10^−4^
Interaction	2	0.0190	0.0095	9.5664	0.0136

**Table 4 materials-13-03989-t004:** Two-way analysis of variance with shrinkage rate.

Source	DF	Sum of Squares	Mean Square	F Value	*p* Value
Factor A (Drying method)	2	18.4091	9.2046	42.0469	0.0003
Factor B (Specimen length)	1	0.4497	0.4497	2.0542	0.2018
Interaction	2	0.1967	0.0979	0.4471	0.0016

**Table 5 materials-13-03989-t005:** Final MC of three drying methods analysis of variance with moisture distribution.

Factor	Drying Method	DF	Sum of Squares	Mean Square	F Value	*p* Value
Final MC	SCD	1	18.0192	18.0192	5.3058	0.0061
	CKD	1	9.0640	9.0640	4.1755	0.7160
	OD	1	4.2689	4.2689	1.0194	0.9020

**Table 6 materials-13-03989-t006:** Two-way analysis of variance with moisture distribution.

Source	DF	Sum of Squares	Mean Square	F Value	*p* Value
Factor A (Drying method)	2	12.9233	6.4617	3.0174	0.0368
Factor B (Specimen length)	1	6.3151	6.3151	5.0654	0.8068
Interaction	2	18.6329	9.3165	5.0964	0.9095

**Table 7 materials-13-03989-t007:** Two-way analysis of variance with drying stress.

Source	DF	Sum of Squares	Mean Square	F Value	*p* Value
Factor A (Drying method)	2	0.3871	0.1935	20.1584	0.0022
Factor B (Specimen length)	1	0.0434	0.0434	4.5241	0.0775
Interaction	2	0.0138	0.0069	0.7177	0.5255

**Table 8 materials-13-03989-t008:** Error analysis data for drying characteristics with drying methods and specimen lengths.

Drying Characteristics	Drying Method	Specimen Length (mm)	Sum of the Squared Errors	Standard Deviation
Drying rate	SCD	50	0.004	0.047
		100	0.001	0.016
	CKD	50	0.001	0.020
		100	0.000	0.003
	OD	50	0.000	0.007
		100	0.000	0.008
Shrinkage	SCD	50	0.001	0.028
		100	0.008	0.065
	CKD	50	0.005	0.048
		100	0.001	0.026
	OD	50	0.004	0.043
		100	0.002	0.030
Drying stress	SCD	50	0.018	0.095
		100	0.015	0.087
	CKD	50	0.010	0.072
		100	0.000	0.014
	OD	50	0.012	0.077
		100	0.001	0.027

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
