# Peer review of "Drying Characteristics of Eucalyptus urophylla × E. grandis with Supercritical CO2"

_materials, 2020, doi:10.3390/ma13183989_

Round 1

Reviewer 1 Report

The paper subject to analysis can be published in Materials. The authors compared the drying process of some different-sized Eucalyptus samples. In addition, they pointed out that the supercritical CO2 drying (SCD) method leads to much better results than the conventional kiln drying (CKD) and oven-drying (OD) methods. However, the authors need to state more clearly in the introductory chapter what the original contribution of their study is.

Reviewer 2 Report

The study is based only on experimental observations, without having a theoretical basis.
Wood drying, in general, is a complex process, influenced by many factors. Wood is an anisotropic material and I do not understand how the rates of moisture migration by radial and tangential directions are not different.
The paper presents comparatively experimental data obtained by three drying techniques. These techniques are not clearly presented, the accuracy of the measurements is not specified and an analysis of the errors is not performed.

Reviewer 3 Report

This is a nice article. It just needs a few tweaks before publication. See below:

L40: Spell out CKD here. I know, it is in the abstract. But, it is written first time in the text here. 

L43: Currently, the CKD method is the most commonly used schedule for eucalyptus industry. 

L44: Drawbacks instead of defects..

L46: Spell SCD out here.

The authors could state the hypothesis similar to: white CKD and OD are the most common drying methods, SCD can be carried to reach high wood drying quality. The overall goal was to improve wood drying procedures for Eucalyptus species. We evaluated drying rate... ... ... to explore the influence of. 

Bottom line intro: It lacks hypothesis, overall goal. But, they authors are not far from there. 

Overall, the Intro was well written. 

Materials and methods

specifications of 30 mm x 25 mm x 1000 mm (transversal x radial, longitudinal).

L65: Very interesting, what was the temperature in the freeze and why use it?

The authors could include a photo of the actual drying system also.

L91: China national Standard - GB/T 1931 (CNS???, 2009)?

l95: Adobe Photoshop version (City, Country). Also, dpi, bits and photo depth would be useful information.

The biggest problem here extends to results is that: How shrinkage was calculate above the fiber saturation point? That directly reflects the results. 

L101-103: At least describe the major points in the standard. Not everybody has access to the standard. 

L102. Moisture content of each block...

Results

L136-138: I think the authors need a reference here. 

L148: "Caused"

Why not make a statistical analysis between the thickness and methods? As a factorial arrangement of treatments in a completely randomized design, where thickness is treatment level A and thickness is treatment B or vice-versa? that would give much more insights about the method. 

Why not show the drying stress samples? That would help to build author's case 

Conclusions are good.

Reviewer 4 Report

“Drying Characteristics of Eucalyptus urophylla × E.grandis with Supercritical CO2” is an interesting piece of work, both from the scientific and practical perspective.

I have a few comments or suggestions to this paper:

  1. Since this is a scientific paper, I would recommend to extend Introduction a little bit and add some more details regarding state-of-the-art. One of the most important advantages of the supercritical CO2 drying method is its ability to sustain the structure of wood and other lignocellulosic materials in an almost unaltered state, despite their chemical composition or degree of degradation, thus limiting its shrinkage and collapse of the cell walls. Therefore, since this is also the main goal of the presented research, I would recommend to develop this part of the Introduction and supplement it with a few more references, showing all the aspects of this issue, e.g. 1. Wang, X., Guo, Y., Zhou, J., & Sun, G. (2017). Structural changes of poplar wood lignin after supercritical pretreatment using carbon dioxide and ethanol-water as co-solvents. RSC advances, 7(14), 8314-8322; 2. Broda, M., Curling, S. F., Spear, M. J., & Hill, C. A. (2019). Effect of methyltrimethoxysilane impregnation on the cell wall porosity and water vapour sorption of archaeological waterlogged oak. Wood Science and Technology, 53(3), 703-726; 3. Dawson, B. S., Pearson, H., Kimberley, M. O., Davy, B., & Dickson, A. R. (2020). Effect of supercritical CO 2 treatment and kiln drying on the collapse in Eucalyptus nitens European Journal of Wood and Wood Products, 1-9 (the last one is mentioned already but in a different part of the paper).
  2. Some minor linguistic corrections are required, e.g. in line 42 “...reduce the drying defects above is to develop…” should be “...reduce the aforementioned drying defects is to develop…” or “…reduce the drying defects mentioned above is to develop…”. The same type of mistake is in line 46.

Another mistake: in line 219: “…because few drying stress occurs…” should be “…because few drying stresses occur

Line 101 – “converts” instead of “convert”.

Line 122 – “considered to be at a state of absolute dryness” or “considered absolutely dry”.

Lines 119-125 – please, use a past tense in all the sentences.

And some more mistakes of these types throughout the text.

  1. “In this paper, we proposed hypothesis that CKD and OD are the most common dying methods, SCD can benefit wood drying quality by comparing CKD and OD.” - I think that the hypothesis should be changed since the part regarding CKD and OD as the most common wood drying methods is not a hypothesis, but a fact. The scientific aim of the research (and the paper) should be clearly specified.
  2. Lines 81-82: 30x25x1000 mm – this is not a square. The word “square” should be removed (the same refers to line 83).
  3. How many samples were used for the particular drying method? How many replicates were used for each measurement?
  4. All the drying steps should be described in the same way – the first three steps are in the form of a command, others are described using past tense (what happened). Sometimes the singular form of “specimen” is used, sometimes plural. Please, unify this part to make it more straightforward.
  5. The described method of the measurement of shrinkage has a drawback – the Authors used the dimensions of a sample which was dried and then re-wetted as the initial dimensions for calculating wood shrinkage. There is a phenomenon called hornification. It shows that, depending on drying methods, changes in wood structure (involving its shrinkage) can be irreversible. Therefore, the assumption made by Authors may be a source of a serious mistake since they applied different drying methods. This part of the experiment should be repeated, using the dimensions of wood samples before and after drying as a basis for shrinkage calculations.
  6. What was the distance of the “moisture distribution slice” from the end/middle part of long and short specimens? Was the distance equal for both lengths of specimens? From Figure 1 it seems that the samples were not taken from the central parts.
  7. How exactly was the drying stress measured? Was it measured on samples before drying OD, CKD or SCD? The description is unclear. Please give some more details.
  8. It would be easier to discuss moisture distribution if the moisture content of samples dried with different methods was equal. The differences in the “moisture pattern” (Figure 5) can result just from the different amount of water inside wood tissue, so it seems to be a little bit tricky to discuss the differences in the MC pattern between samples which have final MC of 51-57%, 117-123% and 81-100%.
  9. Since the research refers only to the initial stage of drying, this fact should be emphasised in the manuscript since the very beginning.
  10. It would be interesting to see the differences in the characteristics of wood dried with different methods to a specific MC (e.g. 50%, but also below FSP).

Round 2

Reviewer 3 Report

I thank the authors for taking the time in making suggestions clearer. This is a really nice manuscript. Congratulations.

Author Response

Thank you very much for your reply.

Reviewer 4 Report

The Authors improved the manuscript and, in my opinion, it meets now the requirements of a scientific paper and the Materials journal. I recommend it for publishing.